# AesMamba:
# Universal Image Aesthetic Assessment with State Space Models

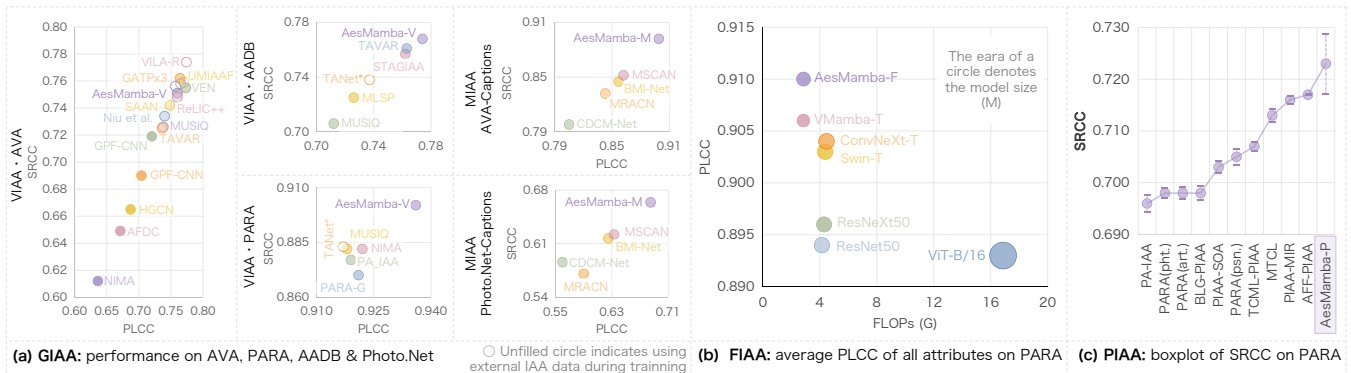

**Figure 1: Our AesMamba models achieve superior or highly competitive performance, in diverse *Image Aesthetic Assessment* (IAA) tasks, across all the benchmark datasets. GIAA, FIAA, and PIAA sequentially indicate the generic, fine-grained, and personalized IAA tasks. Besides, we use VIAA and MIAA to denote visual and multimodal GIAA tasks, respectively. For clarity, the corresponding AesMamba model variants are denoted by -V, -M, -F, and -P accordingly.**

## ABSTRACT

*Image Aesthetic Assessment* (IAA) aims to objectively predict the *generic* or *personalized* evaluations, of the *aesthetic* or *fine-grained multi-attributes*, based on *visual* or *multimodal* inputs. Previously, researchers have designed diverse and specialized methods, for specific IAA tasks, based on different input-output situations. *Is it possible to design a universal IAA framework applicable for the whole IAA task taxonomy?* In this paper, we explore this issue, and propose a modular IAA framework, dubbed AesMamba. Specially, we use the *Visual State Space Model* (VMamba), instead of CNNs or ViTs, to learn comprehensive representations of aesthetic-related attributes; because VMamba can efficiently achieve both global and local effective receptive fields. Afterward, a modal-adaptive module is used to automatically produce the integrated representations, conditioned on the type of input. In the prediction module, we propose a *Multitask Balanced Adaptation* (MBA) module, to boost task-specific features, with emphasis on the tail instances. Finally, we formulate the personalized IAA task as a multimodal learning problem, by converting a user's *anonymous* subject characters to a text prompt. This prompting strategy effectively employs the semantics of flexibly selected characters, for inferring individual preferences. AesMamba can be applied to diverse IAA tasks, through flexible

combination of these modules. Extensive experiments, on numerous benchmark datasets, demonstrate that our AesMamba models consistently achieve superior or highly competitive performance, on all IAA tasks, in comparison with state-of-the-art methods. The code and models will be released after peer review.

## CCS CONCEPTS

• **Computing methodologies → Image representations**.

## KEYWORDS

Image Aesthetic Assessment, State Space Model, Multimodal Learning, Multitask Learning, Imbalanced Learning

## 1 INTRODUCTION

Image Aesthetic Assessment (IAA) aims to prediction the quality of an image, from the aspect of aesthetic. It has a wide range of applications in the image editing, generation, and photographing areas. However, a human's judgement of aesthetic is highly correlated with diverse visual attributes (e.g. color, composition, and content), as well as multiple personal characters [42, 79]. It is still challenging to develop an effective and efficient method, for comprehensively considering all this information during the inference process.

In the past decades, researchers have paid great efforts and have proposed a mass of algorithms. Most of existing works focus on the *Generic IAA* (GIAA) task, which aims to predict the average aesthetic evaluation, assigned by multiple individuals [33, 50, 71]. Recently, researchers start to explore the *Personalized IAA* (PIAA), which learns a specific model for each individual, to predict the personal aesthetic preferences [7]. In addition, several attempts have been made for developing *Fine-grained IAA* (FIAA), i.e. evaluate the quality or subjective preference of multiple visual attributes

*ACM MM, 2024, Melbourne, Australia*
© 2024 Copyright held by the owner/author(s). Publication rights licensed to ACM.
ACM ISBN 978-x-xxxx-xxxx-x/YY/MM
https://doi.org/10.1145/nnnnnnn.nnnnnnn

[23, 31, 79]. In other words, FIAA provides a comprehensive description of aesthetic, instead of a single aesthetic score. Thus, FIAA is of great significance in practical applications.

Existing works typically focus on one single task, and pay efforts to boost the feature representations [33, 89, 92], information fusion mechanism [4, 84], reasoning architectures [27, 45], learning strategies [51, 71, 80, 91], and datasets [1, 24, 57, 79], etc. Inspired by these remarkable progresses, one question arises: *Is it possible to design a universal IAA framework applicable for all these IAA tasks?* To our knowledge, there are mainly the following three challenges.

**Challenge I: Efficient local and global perception.** The subjective judgment of image aesthetic is based on an integration of diverse visual information, from local details (e.g. noise and color) to global perception (e.g. composition and semantic). Thus, both local and global reception fields are imperative for IAA. Advanced IAA methods mainly use *Convolutional Neural Networks* (CNNs) [49, 69] or *Vision Transformers* (ViTs) [12, 48, 72] for learning visual representations. However, neither CNNs nor ViTs can achieve global *Effective Reception Field* (ERF) efficiently [47]. Recently, *State Space Models* (SSMs) [19, 21] have shown superior efficiency in modelling long-range dependence, and have achieved competitive performance in diverse language processing tasks [17] and visual tasks [47, 60]. Inspired by such progress, we explore to use *Vision Mamba* (VMamba) [47] for efficiently learning global visual representations, while preserving local reception field.

**Challenge II: Imbalanced multitask learning.** A universal IAA framework also meets the imbalance learning problem, in both task-level and instance level. *Task-level imbalance.* The prediction of multi-attribute evaluations is naturally a *Multitask Learning* (MTL) problem. Intuitively, the required representations for these attributes, diverse with each other. Besides, the difficulty of learning might vary between different attributes [23, 24, 79]. In the learning process, the attribute evaluation tasks may interfere with each other, scarifying either the stability or effectiveness of representation learning [37]. *Instance-level imbalance.* Besides, in the existing datasets [57, 79], the distribution of aesthetic or attribute scores are heavily imbalanced (Fig. 2). The IAA model would be skewed for lower error, overwhelming the tail labels occupying limited instances [77]. To alleviate the interference among tasks, we propose to adapt the global features to task-specific representations, via *Parameter-Efficient Fine-Tuning* (PEFT) [11, 56]. Besides, we use an auxiliary scale categorization task, and optimize it using the *Balanced Cross-Entropy* (Bal-CE) loss [77], to strengthen the contributions of tail labels. The whole solution, termed *Multitask Balanced Adaptation* (MBA), will be detailed in Section 3.4.

**Challenge III: Flexible design.** To enable a universal IAA framework, it's necessary to design a unified and flexible pipeline, which is applicable in all situations. *Flexible PIAA pipeline.* Although there have been numerous and diverse PIAA methods [42, 52, 61], they neglect the semantic information of subject characters. Besides, it's difficult to flexibly modify the characters, in the recent conditional PIAA method [79]. To boost the flexibility and precision of PIAA, in this paper, we propose to convert a user's multiple subject characters to a text prompt; and then predict personalized evaluations based on both the image and the text prompt (Section 3.5). In this way, PIAA is formulated as a multimodal learning task. Besides, the text prompt allows flexible combinations of

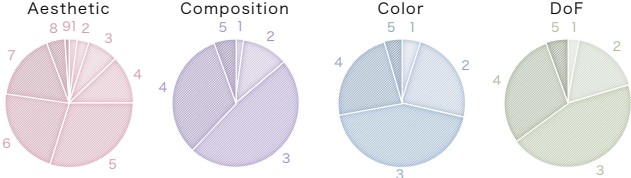

**Figure 2: Pie charts of the imbalanced distributions of *discritized* aesthetic and attribute scores, in PARA [79].**

arbitrary subject characters. ***Flexible inputs and outputs.*** In IAA tasks, the input might be merely an image (i.e. *visual*) or an image-text pair (i.e. *multimodal*); and the output might be the aesthetic evaluations (i.e. *single-task*) or multi-attribute labels (i.e. *multitask*). We thus use a *Modal-Adaptive Integration* (MAI) mechanism, to automatically integrate the input visual or multimodal features, inspired by the *Mixtures-of-Modal-Experts* (MoME) [2].

Based on all the above considerations, we propose a universal IAA framework, dubbed AesMamba, with modular design. Specifically, AesMamba mainly includes: (1) a group of image-text encoders, (2) the MAI module for feature integration, and (3) the MBA module for feature adaptation and balanced aesthetic/attribute evaluations. With a flexible combination of these modules, Aesmba is applicable to different IAA tasks. We conduct extensive experiments on diverse IAA tasks, across several IAA datasets, including AVA [57], TAD66K [24], PARA [79], AADB [35], and Photo.Net [8]. As shown in Fig. 1, AesMamba models show superior or at least competitive performance across all the tasks, in comparison with state-of-the-art (SOTA) IAA models or advanced visual backbones. We also conducted a series of ablation study, to verify the effectiveness of the proposed techniques.

Our contributions in this paper are summarized as follows:

- First, we propose a universal IAA framework, AesMamba, with modular design. AesMamba is applicable to all the IAA tasks, by flexible combining the proposed modules.
- We propose to use Visual Mamba for efficiently achieving both global and local ERFs. To our best knowledge, this is the first use of SSMs in the IAA area.
- We propose a novel PIAA pipeline, via *Subject Multi-character Prompts* (SMP) and multimodal learning. The semantic information of multiple subject characters are taken into consideration in the inference stage.
- To combat the imbalance problems in IAA, we propose a *Multitask Balanced Adaptation* (MBA), to learn task-adaptive representations with emphasis on the tail instances.
- Our AesMamba models achieve superior or highly competitive performance on all IAA tasks, across diverse datasets. The code and models will be released after peer review.

## 2 RELATED WORKS

**Generic IAA (GIAA).** Existing IAA works mainly focus on *Visual GIAA* (**VIAA**), which predicts the average aesthetic evaluation of human users by learning visual representations from a given image. Traditional methods use manual features to describe different elements of visual aesthetics [10, 38], such as layout, content and

lighting. With the development of deep learning, advanced methods usually employ CNNs [69, 71] or Transformers [6, 12, 32, 48, 72] for aesthetic representation learning. Besides, researches propose to use multiscale features [29], semantic [26], attributes [43], structure [89], emotion [5, 36], knowledge embedding [41] and other auxiliary information to enhance aesthetic representation. In recent years, researchers have begun to study *multimodal GIAA* (**MIAA**), which uses textual comments as an auxiliary input from which to mine high-level semantic information about human aesthetic judgments [59, 84]. Moreover, inspired by visual-language representation learning [63] and *Large-scale Language Models* (LLMs) [3], researchers have explored the possibility of learning language representation based on large-scale pre-training [33, 61, 65], description generation [90] and evaluation framework via LLMs [30]. However, these works require a large amount of pre-trained data and huge computational resources.

**Personalized IAA (PIAA).** In addition, researchers have explored a variety of PIAA methods [94] based on aesthetic elements and user characters [42, 45, 93] to predict individually aesthetic preferences. Existing PIAA methods typically try to transfer a pre-trained GIAA model to a specific user, based on collaborative filtering [7, 73], user interactions [27, 52, 95], or preference divergence [64, 92]. More recently, researchers have explored multitask learning [42], meta learning [78, 91], contrast learning [80], and reinforcement learning [51] for better discovering users' preferences, and to use federated learning [76] for privacy protection. Besides, researchers constructed several PIAA datasets, e.g. FLICKR-AES [64], and PARA [79], and PR-AADB [16]. However, existing methods typically use CNNs [92] or Transformers [78] for learning attribute-aware features, and don't allow flexible change of the conditioned subject characters [79].

**Fine-grained IAA (FIAA).** Fine-grained IAA (FIAA) aims to jointly evaluate the aesthetic quality from multiple aspects. Note that previous multitask or multibranch IAA methods [24, 36] only predict the category of an attribute. In contrast, FIAA requires to predict a score for each aesthetic-related attribute. For now, several attempts have been made for developing FIAA algorithms. For example, Jin et al. [31] constructed the AMD-A dataset with three attribute annotations, and use a combination of deep and handcrafted features for prediction. Soydaner et al. [70] use a naive multitask CNN for this task. Recently, He et al. propose to evaluate image color aesthetic via Transformers and construct a specific dataset [23]. Besides, PARA [79] includes multi-attribute scores and can also serve as a FIAA benchmark.

**State Space Models (SSMs).** State Space Models (SSMs) are recently proposed models for solving long-range dependency problem [18, 19, 21], and have shown inspiring performance in *Natural Language Processing* (NLP) [55] and sequence reasoning [17]. Recently, esearchers start to explore Mamba models for visual tasks [47, 96]. For example, Liu et al. [47] proposed VMamba based on the *Selective State Space Models* (Mamba) [17], by designing a *Cross-Scan Module* (CSM) for efficiently processing images. Besides, researchers propose to optimize the selective scanning strategies [28, 60] or the attention mechanism [96], to further boost the efficiency. Along with such progresses, Mamba models have been applied to image segmentation, or multimodal large language models [62].

## 3 AESMAMBA

### 3.1 Overview

In the taxonomy of IAA tasks, the input might be merely an image (i.e. *visual*) or an image-text pair (i.e. *multimodal*); and the output might be merely an aesthetic label (i.e. *single-task*) or multi-attribute labels (i.e. *multitask*). To enable a universal IAA framework, applicable to all of these situations, we modularize the IAA framework, and proposed AesMamba. As shown in Fig. 3, AesMamba mainly includes the following modules: (1) first, the multimodal encoders efficiently transform the input image and text to effective representations (Section 3.2); (2) second, the *Modal-Adaptive Integration* (MAI) mechanism automatically produces integrated global features, conditioned on the modal of input (Section 3.3); (3) finally, the *Multi-Balanced Adaptation* (MBA) adapts the global feature to specific prediction tasks, and outputs the estimated labels. In MBA, we propose to use an auxiliary balanced categorization branch to each task, to tackle the challenge of imbalanced learning (Section 3.4). Besides, we formulate PIAA as a multimodal task, by transferring a user's personal information into a prompt of subject characters (Section 3.5). As shown in Fig. 3, AesMamba can be applied to different IAA tasks, with flexible combinations of these modules. For clarity, we refer to the model variants as AesMamba-$\alpha$, where $\alpha$ is the first letter of the corresponding IAA task, i.e. VIAA, MIAA, FIAA, or PIAA. Details of each module and the corresponding variants are presented below.

### 3.2 multimodal Encoders

In IAA tasks, the input might be merely an image (i.e. visual) or an image-text pair (i.e. multimodal). In AesMamba, we use VMamba [47] as the image encoder, for efficiently gaining both local and global *Effective Reception Field* (ERF). Besides, we use BERT [9] as the text encoder, to transform the text comments or prompts of subject characters into high-dimensional representations.

*3.2.1 Image Encoder: VMamba.* In the implementation, we use the tiny version of VMamba (VMamba-T) in default, unless otherwise specified. Fig. 4(a) shows the overall pipeline of VMamba-T. Given an input image, VMamba first divides it into multiple patches, generating a feature map (*Stem*). Afterward, a stack of *Visual State Space* (VSS) blocks are used to hierarchically process and downsample the feature map, thorough 4 stages. The final output of the last stage is adopted as the visual representations, i.e. $\mathbf{F}_v$.

**VSS Block.** The structure of the VSS block is as shown in Fig. 4(b). The input feature is first undergone a *Layer Normalization* (LN) layer, and then processed through two streams. The major stream includes a linear layer, a *Depth-wise Convolutional* (DW-Conv) layer, a SiLU activation function [66], a *2D Selective Scan* (SS2D) mechanism, and an LN layer, sequentially. The other stream includes a linear layer, followed by the SiLU activation. Finally, the element-wise product of these two branch outputs is fed into a linear layer, and then added with the original input.

**2D Selective Scan (SS2D).** Fig. 4(c) shows the pipeline of the SS2D mechanism. In the *scan expand* stage, the feature map is scanned in each of the four directions and divided into four sequences. Then, the four sequences are processed separately, through the *selective scanning* (**S6**) block [17], for capturing comprehensive

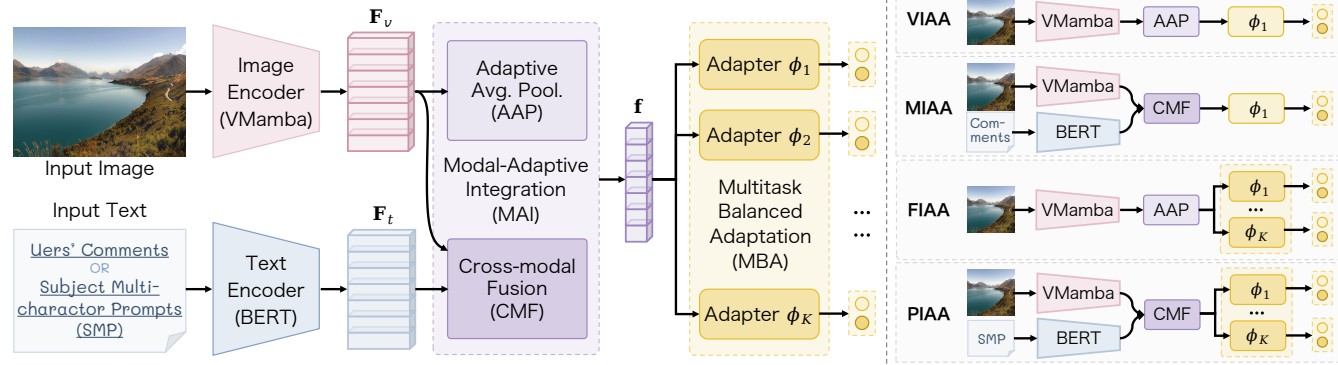

**Figure 3: Overview of AesMamba with modular design (*left*), and its applications in different IAA tasks (*right*). The *Modal-adaptive Integration* (MAI) mechanism automatically switches between *Adaptive Average Pooling* (AAP) and *Cross-modal Fusion* (CMF), based on whether the input is visual (VIAA & FIAA) or multimodal (MIAA & PIAA). *Multitask Balanced Adaptation* (MBA) can be used for either aesthetic prediction (*single-task*) or multi-attribute evaluation (*multitask*). We formulate PIAA as a multimodal learning task, by converting user's (*anonymous*) attributes to the *Subject Multi-character Prompts* (SMP).**

and diverse representations [47]. Let $x \in \mathbb{R}^{B \times L \times D}$ denotes an input sequence, and $x_k \in \mathbb{R}^{B \times L \times D}$ a sampled vector within it. Here, $B$ is the batch size, $L$ is the length of this sequence, and $D$ is the dimension of features. $x_k$ is mapped to $y_k \in \mathbb{R}^{B \times L \times D}$ through a hidden state $h_k \in \mathbb{R}^{B \times L \times N}$, with the evolution parameter $\mathbf{A} \in \mathbb{R}^{N \times N}$ and the projection parameters $\mathbf{B} \in \mathbb{R}^{B \times L \times N}$, $\mathbf{C} \in \mathbb{R}^{B \times L \times N}$, i.e.

$$h_k = \bar{\mathbf{A}}h_{k-1} + \bar{\mathbf{B}}x_k,$$
$$y_k = \mathbf{C}h_k + x_k, \tag{1}$$
$$\text{with } \bar{\mathbf{A}} = e^{\Delta\mathbf{A}}, \quad \bar{\mathbf{B}} = (e^{\Delta\mathbf{A}} - \mathbf{I})\mathbf{A}^{-1}\mathbf{B},$$

where $(\bar{\mathbf{A}}, \bar{\mathbf{B}})$ is the discretized version of $(\mathbf{A}, \mathbf{B})$ through *Zero-Order Hold* (ZOH) [20, 21]; $\Delta \in \mathbb{R}^{B \times L \times N}$ is the timescale parameter. In the implementation, $\mathbf{B}$, $\mathbf{C}$, and $\Delta$ are derived from the input data $x$, through linear layers. Following [17], the approximation of $\bar{\mathbf{B}}$ is refined by the first-order Taylor series, i.e. $\bar{\mathbf{B}} = \Delta\mathbf{B}$. Finally, each processed sequence is reshaped into a feature map, and all the four feature maps are merged to form a new one, in the *scan merge* stage.

*3.2.2 Text Encoder: BERT.* BERT [9] is based on the Transformer architecture, and has been well pre-trained on large language corpora. Besides, it has proven very efficient and effective in transferring to various tasks. We therefore use BERT to transform the text to high-dimensional representations $\mathbf{F}_t$. In the IAA scenario, the input text is the users' comment or the given prompt. We first divide the text into a word sequence $\{w_1, ..., w_n\}$ of length $n$, using the WordPiece Tokenizer [74]. Afterward, a [cls] token is added at the beginning of this sequence, and is optimized to be the final textual representation. The text encoder is initialized with weights pre-trained on the wikipedia and bookcorpus corpora, and fine-tuned simultaneously with the aesthetic prediction task.

### 3.3 Modal-adaptive Feature Integration (MAI)

To enable our model applicable in both situations, we use a *Modal-adaptive Integration* (MAI) mechanism, to automatically integrate the encoded features to an integrated representation.

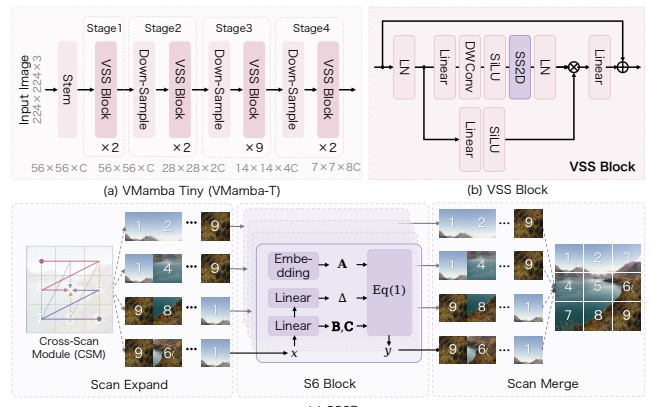

**Figure 4: Architectures of (a) our visual backbone, i.e. VMamba-T, (b) the VSS block, and (c) the SS2D process [47].**

**Adaptive Average Pooling (AAP).** In the scenario of merely visual input, an input image is transformed to its visual features $\mathbf{F}_v \in \mathbb{R}^{m \times d}$ by the image encoder. To obtain an integrated representation, we pool $\mathbf{F}_v$ to a global feature vector, i.e. $\mathbf{f}_v \in \mathbb{R}^{1 \times d}$, via *Adaptive Average Pooling* (AAP). In this case, the output of MAI is the same as $\mathbf{f}_v$, i.e. $\mathbf{f} = \mathbf{f}_v$, and will be used for quality prediction.

**Cross-modal Fusion (CMF).** In the multimodal scenario, we use a *Cross-modal Fusion* (CMF) modal (Fig. 5) to integrate image-text features. The key of CMF is *Cross-Attention* (CA). Specifically, the global visual feature $\mathbf{f}_v \in \mathbb{R}^{1 \times d}$ is linearly transformed to a query $\mathbf{Q} \in \mathbb{R}^{1 \times d}$, and the text feature $\mathbf{F}_t \in \mathbb{R}^{n \times d}$ is transformed to the key $\mathbf{K} \in \mathbb{R}^{n \times d}$ and value $\mathbf{V} \in \mathbb{R}^{n \times d}$, through a linear layer, respectively. The computation of CA is formulated as:

$$\text{CA}(\mathbf{f}_v, \mathbf{F}_t) = \text{softmax}\left(\frac{\mathbf{Q}\mathbf{K}^\top}{\sqrt{d}}\right)\mathbf{V} = \mathbf{A}\mathbf{V}, \tag{2}$$

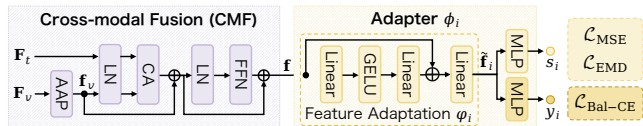

**Figure 5: Pipelines of *Cross-modal Fusion* (CMF) and the adapter in *Multitask Balanced Adaptation* (MBA).**

Besides, the *Layer Normalization* (LN) and *Feedforward Neural Network* (FFN) are used in CMF to boost the fused feature. The whole fusion process can be expressed as:

$$\tilde{\mathbf{f}}_v = \mathbf{f}_v + \text{CA}(\text{LN}(\mathbf{f}_v), \text{LN}(\mathbf{F}_t)),$$
$$\mathbf{f} = \tilde{\mathbf{f}}_v + \text{FFN}(\text{LN}(\tilde{\mathbf{f}}_v)). \tag{3}$$

Finally, the integrated feature $\mathbf{f} = \text{CMF}(\mathbf{f}_v, \mathbf{F}_t)$ is fed into subsequent modules for aesthetic prediction.

## 3.4 Multitask Balanced Adaptation (MBA)

Fine-grained aesthetic evaluation is a typical *Multitask Learning* (MTL) problem. If we extend several MLPs directly after MAI, different tasks may interfere with each other and affect the effectiveness and stability of the integrated feature learning [37]. To address this challenge, we propose a *Multitask Balanced Adaptation* (MBA) module. Specifically, for each attribute prediction task, an independent lightweight adapter module $\phi_i, i = 1, 2, ..., K$ is expanded after the global features. The adapter consists of a feature adaptation module and an MLP for aesthetic prediction.

*3.4.1 Feature Adaptation.* The feature adaptation module includes two FC layers, with a GELU activation function and a residual connection. Given the previously processed feature $\mathbf{f} \in \mathbb{R}^{1 \times d}$, the *feature adaptation* process of the $i$-th task is formulated as:

$$\tilde{\mathbf{f}}_i = \varphi_i(\mathbf{f}) = \text{Linear}_2(\text{GELU}(\text{Linear}_1(\mathbf{f}))) + \mathbf{f}. \tag{4}$$

The first linear layer reduces the feature dimension to $D/4$, while the second one increases the dimension back to $D$. As a result, a specific feature space is learned for each task, avoiding the problem of feature interference and confusion in multitask learning. Finally, $\tilde{\mathbf{f}}_i$ is fed into the MLP for predicting the $i$-th attribute.

*3.4.2 Imbalanced Scale Categorization.* To address the challenge of imbalanced data distribution, we use an auxiliary categorical aesthetic assessment task in each branch *during training*. To this end, we discretize a continuous attribute score $s_i$ to a scale category $y_i$. Besides, we use another MLP branch to predict the attribute scale category based on $\tilde{\mathbf{f}}_i$. During training, we use the *Balanced Cross-Entropy* (Bal-CE) loss [77] between the predicted scale label $\hat{y}_i$ and the ground-truth label $y_i$ to jointly optimize the whole network. Specifically, the Bal-CE loss considers the number of instances of each class $n_{y_i}$ in loss computation, i.e.

$$\mathcal{L}_{\text{Bal-CE}} = \sum_{i=1}^{K} -\log\left[\frac{n_{y_i} e^{z_{i,y_i}}}{\sum_{l=1}^{C_i} n_l e^{z_{i,l}}}\right], \tag{5}$$

where, $n_l \in \mathcal{Y}_i$ is the number of instances with true label $l \in \{1, \dots, C_i\}$ in class $i$; $K$ is the number of tasks. The Bal-CE loss improves the significance of rare scale categories in the learning process. As a result, this auxiliary scale categorization task might improve the model's ability to correctly evaluate long-tail data.

## 3.5 Subject Multi-character Prompts (SMP)

PIAA aims to learn a specific IAA model for each user, for predicting his/her aesthetic preferences, conditioned on subject characters. As proved in previous works [79], diverse subject characters, e.g. the artistic and photographic experience, are correlated with personal aesthetic preferences. However, previous PIAA methods can only take one single character as input. To combat this challenge, we propose to transfer users' character labels to *Subject Multi-character Prompts* (SMP), by designing a text template.

**Subject Multi-character Prompts (SMP).** In this paper, we consider the three user information provided in the PARA dataset [79]: artistic experience, photographic experience, and personality traits. Specifically, the aesthetic experience and photographic experience both contain four levels: "*beginner*", "*competent*", "*proficient*", and "*expert*". To incorporate this information into the model, we design the corresponding text prompts as:

> "My artistic experience is {art_exp}."

and

> "My Photographic experience is {photo_exp}."

where `art_exp` and `photo_exp` represent the corresponding experience levels of a user. The personality traits consist of five aspects, i.e. "*Conscientiousness*", "*Agreeableness*", "*Extroversion*", "*Openness*", and "*Neuroticism*". Given a user, let $C$, $A$, $E$, $O$, and $N$ denote the corresponding trait scores sequentially. The corresponding text prompts follow the template below:

> "In the Big-Five personality traits test, my scores are as follows: openness score is {O}, conscientiousness score is {C}, extroversion score is {E}, agreeableness score is {A}, and Neuroticism score is {N}."

All the above prompts can be integrated together, to represent the personal information of users, i.e. $\mathcal{I}_u$. Besides, our MCP method allows great flexibility in attribute selection and extension.

**AesMamba-P.** As shown in Fig. 3, AesMamba-P takes an image and the text prompts as inputs, and then encoding them to deep features $\mathbf{F}_v$ and $\mathbf{F}_t$, through the image/text encoders, respectively. $\mathbf{F}_t$ comprehensively represents personal information of a user. Afterward, $\mathbf{F}_v$ and $\mathbf{F}_t$ are fused in the CMF module (Eq. 3) and fed to the MBA module for personalized aesthetic prediction.

## 3.6 Loss functions

In some benchmark GIAA datasets [57], the score distribution of each image is available. In this case, we have our model predicting the aesthetic score distributions, and use the *Earth Mover's Divergence* (EMD) [46, 71] loss, i.e.

$$\mathcal{L}_{\text{EMD}} = \left(\frac{1}{10}\sum_{i=1}^{10}(|\text{CDF}(\mathbf{a}) - \text{CDF}(\mathbf{p})|)^r\right)^{\frac{1}{r}}, \tag{6}$$

where $\mathbf{a}$ and $\mathbf{p}$ are the predicted distribution and the ground truth, respectively; CDF() is the *Cumulative Distribution Function* (CDF); $r$ is set to 2 during training. In the inference stage, an aesthetic score is computed based on the predicted distribution [71]. In other

datasets [79], only the average scores are available. In this case, we use the *Mean Squared Error* (MSE) loss, i.e.

$$\mathcal{L}_{\text{MSE}} = \frac{1}{n} \sum_{i=1}^{n} (s_i - \hat{s}_i)^2 \qquad (7)$$

where $n$ is the number of training samples.

Finally, we add the Bal-CE loss (Eq. 6) to the EMD or MSE loss, as the full objective. The total loss for the *score distribution prediction* task, $\mathcal{L}_{dst}$, or the *score regression* task, $\mathcal{L}_{reg}$, are computed by:

$$\mathcal{L}_{dst} = \mathcal{L}_{\text{EMD}} + \gamma \mathcal{L}_{\text{MSE}} + \lambda \mathcal{L}_{\text{Bal-CE}},$$
$$\text{or } \mathcal{L}_{reg} = \mathcal{L}_{\text{MSE}} + \lambda \mathcal{L}_{\text{Bal-CE}}, \qquad (8)$$

where $\gamma$ and $\lambda$ are weighting factors and set to 10 and 0.001, respectively, in the implementation.

## 4 EXPERIMENTS

### 4.1 Settings

*4.1.1 Datasets.* We conduct experiments on the following benchmark datasets, i.e. the AVA [57], TAD66K [24], PARA [79], AADB [35], and Photo.Net [8] datasets, which sequentially contains about 255K, 66K, 20K, 31K, and 10K images. Both AVA and Photo.Net include the aesthetic score distribution of each image; while PARA and AADB include the average score. We conducted VIAA experiments on all these datasets separately (Section 4.2.1). Besides, for both AVA and Photo.Net, we crawl valid comments and conduct MIAA experiments on the corresponding expanded datasets, i.e. AVA-Captions [14] and Photo.Net-Captions (Section 4.2.2). Finally, PARA [79] consists of both generic and personalized annotations of multiple aesthetic attributes. We thus conduct FIAA and PIAA experiments on PARA (Sections 4.3 and 4.4). We adopt the same division of each dataset as previous works [79, 84].

During training, an image is resized to $256 \times 256$, followed by random cropping to obtain patches of $224 \times 224$. Random horizontal flipping is applied with a probability of 0.5 for data augmentation. In the test stage, an image is resized to $224 \times 224$, and then fed into the learned model for aesthetic/attributes prediction.

*4.1.2 Criteria.* We use three performance indices as the criteria, i.e. the aesthetic classification accuracy (*Acc.*); the *Pearson's Linear Correlation Coefficient* (PLCC), and the *Spearman Rank-order Correlation Coefficient* (SRCC) between subjective scores and predicted ones. Higher values of these criteria indicate better performance.

*4.1.3 Implementation Details.* In all experiments, we use AdamW optimizer [34] with *Stochastic Gradient Descent* (SGD) during training. Momentum parameters $\beta_1$ and $\beta_1$ are set to 0.9 and 0.99, respectively. In the PIAA task, the initial learning rate is set to $4 \times 10^{-5}$, with a batch size of 10/50 under the 10shot/100shot settings, respectively. In the other IAA tasks, the initial learning rate is set to $1 \times 10^{-4}$, with a batch size of 64. The cosine annealing algorithm is also used to dynamically adjust the learning rate. The lowest learning rate threshold is set to $1 \times 10^{-6}$. All models are implemented using the PyTorch framework, and trained/test on an NVIDIA GeForce RTX3090 with 24GB of memory.

**Table 1: Visual GIAA (VIAA) performance on AVA, TAD66K, PARA, AADB, and Photo.Net. * indicates using external IAA data during training. † indicates that VMamba-B is used as the image encoder in our AesMamba-V.**

(a) AVA

| AVA | Acc. | PLCC | SRCC |
|---|---|---|---|
| MNA-CNN[54] | 76.10 | - | - |
| A-Lamp[53] | 82.5 | - | - |
| NIMA[71] | 81.5 | 0.636 | 0.612 |
| MUSIQ[32] | 81.5 | 0.738 | 0.726 |
| MLSP*[25] | 81.7 | 0.757 | 0.756 |
| ReLIC++[89] | 82.4 | 0.760 | 0.748 |
| AFDC[6] | 83.2 | 0.671 | 0.649 |
| UIAA*[83] | 80.8 | 0.720 | 0.719 |
| SAAN[81] | 80.6 | 0.748 | 0.742 |
| HGCN[67] | 84.6 | 0.687 | 0.665 |
| UMIAAF*[44] | 81.7 | 0.770 | 0.759 |
| VEN[87] | 83.6 | 0.773 | 0.755 |
| GPF-CNN[85] | 81.8 | 0.704 | 0.690 |
| TAVAR[40] | - | 0.736 | 0.725 |
| GATP$_{\times 3}$[15] | - | 0.764 | 0.762 |
| PA_IAA*[42] | 83.7 | - | 0.677 |
| TANet*[24] | - | 0.765 | 0.758 |
| VILA-R*[33] | - | **0.774** | **0.774** |
| AesMamba-V | **84.6** | 0.760 | 0.751 |

(c) PARA

| PARA | Acc. | PLCC | SRCC |
|---|---|---|---|
| PA_IAA*[42] | 87.5 | 0.919 | 0.877 |
| NIMA[71] | 89.0 | 0.922 | 0.882 |
| MUSIQ[32] | 88.1 | 0.918 | 0.882 |
| TANet*[24] | **89.2** | 0.917 | 0.883 |
| PARA-G[79] | 87.0 | 0.921 | 0.879 |
| AesMamba-V | 88.7 | **0.936** | **0.902** |

(b) TAD66K

| TAD66K | Acc. | PLCC | SRCC |
|---|---|---|---|
| A-Lamp[53] | - | 0.422 | 0.411 |
| NIMA[71] | - | 0.405 | 0.390 |
| AADB*[35] | - | 0.400 | 0.379 |
| BIAA*[91] | - | 0.431 | 0.417 |
| PAM*[64] | - | 0.440 | 0.422 |
| UIAA*[82] | - | 0.441 | 0.433 |
| MP$_{ada}$[68] | - | 0.480 | 0.466 |
| HGCN*[67] | - | 0.493 | 0.486 |
| MLSP*[25] | - | 0.508 | 0.490 |
| TANet*[24] | - | **0.531** | **0.513** |
| AesMamba-V | 67.3 | 0.503 | 0.475 |
| AesMamba-V† | **72.0** | 0.511 | 0.483 |

(d) AADB

| AADB | Acc. | PLCC | SRCC |
|---|---|---|---|
| UIAA*[82] | - | - | 0.726 |
| HIAA[39] | - | - | 0.739 |
| MUSIQ[32] | 76.3 | 0.712 | 0.706 |
| TANet*[24] | 79.8 | 0.737 | 0.738 |
| MLSP*[25] | 78.2 | 0.726 | 0.725 |
| STAGIAA[4] | 81.6 | 0.762 | 0.757 |
| TAVAR[40] | 81.9 | 0.763 | 0.761 |
| AesMamba-V | **82.9** | **0.774** | **0.768** |

(e) Photo.Net

| Photo.Net | Acc. | PLCC | SRCC |
|---|---|---|---|
| GPF-CNN[85] | 75.6 | 0.546 | **0.522** |
| AesMamba-V | **80.3** | **0.547** | 0.518 |

## 4.2 Comparison with SOTAs in GIAA

We first compare our method with a mass of advanced methods in the *Visual GIAA* (VIAA) and *multimodal GIAA* (MIAA) tasks.

*4.2.1 VIAA Performance.* Table 1 shows the VIAA performance of AesMamba-V and existing algorithms. Note that some previous methods (denoted by *), e.g. TANet [24] and VILA-R [33], use external aesthetic-related datasets (e.g. FLICKR-AES [64] and LAION-5B [65]) during training or pre-training. On contrast, all the other methods only use the standard training set on each dataset, separately. Across all these datasets, AesMamba-V achieves optimal or highly competitive performance, compared to a mass of advanced methods. ESpecifically, AesMamba-V outperforms all the existing methods, including TANet or MLSP, by 0.01-0.02 percents in PLCC/SRCC, on both PARA and AADB. Besides, AesMamba-V becomes highly competitive with MLSP on TAD66K, when we use VMamba-B (instead of VMamba-T) as the image encoder (denoted by AesMamba-V†). Such observation implies the potential of boosting AesMamba-V by using more advanced ViM models. The outstanding performance of VILA-R implies the significance of large-scale pretraining. However, the required huge computational burden is a grand challenge in

**Table 2: multimodal GIAA (i.e. MIAA) performance on the AVA-Captions and Photo.Net-Captions datasets.**

|  | AVA-Captions | | | Photo.Net-Captions | | |
|---|---|---|---|---|---|---|
|  | Acc. | PLCC | SRCC | Acc. | PLCC | SRCC |
| MRACNN[86] | 85.7 | 0.843 | 0.832 | 78.9 | 0.590 | 0.571 |
| MSCAN[84] | 86.7 | 0.862 | 0.852 | 81.0 | 0.625 | 0.617 |
| BMI-Net[58] | 86.5 | 0.857 | 0.845 | 80.4 | 0.633 | 0.622 |
| CDCM-Net[88] | 86.6 | 0.805 | 0.798 | 81.7 | 0.560 | 0.586 |
| AesMamba-M | **89.6** | **0.899** | **0.892** | **82.0** | **0.685** | **0.664** |

**Table 3: FIAA performance on the PARA dataset.**

|  | Aesthetic | | Quality | | Composition | | Color | | DoF | |
|---|---|---|---|---|---|---|---|---|---|---|
|  | PLCC | SRCC | PLCC | SRCC | PLCC | SRCC | PLCC | SRCC | PLCC | SRCC |
| ResNet50[22] | 0.919 | 0.873 | 0.924 | 0.872 | 0.883 | 0.836 | 0.888 | 0.859 | 0.897 | 0.855 |
| ResNeXt50[75] | 0.922 | 0.880 | 0.924 | 0.875 | 0.892 | 0.852 | 0.889 | 0.862 | 0.883 | 0.840 |
| ViT-B/16[13] | 0.917 | 0.877 | 0.921 | 0.876 | 0.887 | 0.846 | 0.888 | 0.865 | 0.892 | 0.851 |
| Swin-T[48] | 0.926 | 0.888 | 0.930 | 0.886 | 0.895 | 0.855 | 0.896 | 0.871 | 0.905 | 0.866 |
| ConvNeXt-T[49] | 0.928 | 0.890 | 0.931 | 0.887 | 0.898 | 0.858 | 0.896 | 0.870 | 0.905 | 0.868 |
| VMamba-T[47] | 0.929 | 0.896 | 0.934 | **0.894** | 0.901 | 0.865 | 0.897 | 0.872 | 0.908 | 0.872 |
| AesMamba-F | **0.934** | **0.898** | **0.936** | **0.894** | **0.904** | **0.867** | **0.903** | **0.877** | **0.912** | **0.877** |

|  | Light | | Content | | Preference | | Share | | Average | |
|---|---|---|---|---|---|---|---|---|---|---|
|  | PLCC | SRCC | PLCC | SRCC | PLCC | SRCC | PLCC | SRCC | PLCC | SRCC |
| ResNet50[22] | 0.891 | 0.848 | 0.882 | 0.830 | 0.886 | 0.852 | 0.880 | 0.849 | 0.894 | 0.853 |
| ResNeXt50[75] | 0.875 | 0.833 | 0.863 | 0.810 | 0.872 | 0.838 | 0.880 | 0.850 | 0.896 | 0.857 |
| ViT-B/16[13] | 0.891 | 0.856 | 0.881 | 0.832 | 0.884 | 0.851 | 0.878 | 0.848 | 0.893 | 0.856 |
| Swin-T[48] | 0.900 | 0.864 | 0.892 | 0.846 | 0.893 | 0.860 | 0.888 | 0.857 | 0.903 | 0.866 |
| ConvNeXt-T[49] | 0.900 | 0.862 | 0.890 | 0.843 | 0.895 | 0.862 | 0.889 | 0.860 | 0.904 | 0.867 |
| VMamba-T[47] | 0.902 | 0.867 | 0.894 | 0.853 | 0.899 | 0.870 | 0.892 | 0.867 | 0.906 | 0.873 |
| AesMamba-F | **0.905** | **0.868** | **0.900** | **0.854** | **0.901** | **0.871** | **0.895** | **0.869** | **0.910** | **0.875** |

most scenarios. In contrast, AesMamba-V is lightweight (with about 20M parameters) and can be optimized in about one hour on AVA.

*4.2.2 MIAA Performance.* Table 2 shows that our AesMamba-M achieves the best performance in terms of all the indices. Compared to previous SOTAs, AesMamba-M achieves 2.9, 0.037, and 0.040 points *absolute* improvements in Accuracy, PLCC, and SRCC, respectively, on AVA-Captions; as well as over 0.05/0.04 points *absolute* improvements in PLCC/SRCC on Photo.Net-Captions. Besides, AesMamba-M distinctly outperforms all the VIAA methods (Table 1), including VILA-R and AesMamba-V, on both datasets. Such distinct superiority demonstrates the crucial role of text comments in IAA, as well as the effectiveness of our *Cross-modal Fusion* (CMF) module in integrating vision-language representations.

## 4.3 Comparison with SOTAs in FIAA

We further evaluate our AesMamba in the *Fine-grained* IAA (FIAA) task on PARA. We here use the corresponding AesMamba-F to simultaneously predict all the aesthetic and attribute scores (Fig. 2). Since none of the existing IAA methods have officially reported their FIAA performance, we compare our model with a number of advanced networks, including our visual backbone, VMamba-T. All the models are initiated with officially released parameters, and fine-tuned with the MSE loss (Eq. 7) on the standard training set.

Table 3 shows the PLCC and SRCC about every attribute, as well as the average values across all the attributes. Obviously, our model consistently achieves the best PLCC and SRCC values, across all the

**Table 4: PIAA performance on the PARA dataset.**

|  | PLCC | | SRCC | |
|---|---|---|---|---|
|  | 10shot | 100shot | 10shot | 100shot |
| PARA-*art.*[79] | 0.733±0.0022 | 0.742±0.0012 | 0.686±0.0016 | 0.698±0.0012 |
| PARA-*pht.*[79] | 0.733±0.0010 | 0.745±0.0010 | 0.683±0.0014 | 0.698±0.0010 |
| PARA-*psn.*[79] | 0.738±0.0007 | 0.750±0.0010 | 0.691±0.0009 | 0.705±0.0015 |
| TCML$_{PIAA}$[78] | - | - | 0.700±0.0007 | 0.707±0.0009 |
| MTCL[80] | - | - | 0.695±0.0011 | 0.713±0.0013 |
| BLG-PIAA[91] | - | - | 0.688±0.0015 | 0.698±0.0013 |
| PA_IAA[42] | - | - | 0.683±0.0013 | 0.696±0.0016 |
| PIAA-MIR[95] | - | - | 0.702±0.0010 | 0.716±0.0008 |
| PIAA-SOA[93] | - | - | 0.690±0.0014 | 0.703±0.0012 |
| AFF-PIAA[92] | - | - | 0.704±0.0010 | 0.717±0.0010 |
| AesMamba-P | **0.749±0.0083** | **0.763±0.0087** | **0.707±0.0049** | **0.723±0.0058** |

attributes. Besides, our base model, i.e. VMamba-T, consistently outperforms all the other advanced visual backbones, either CNNs or Transformers. Such stable superiority demonstrate our motivation of using VMamba for learning effective aesthetic representations.

## 4.4 Comparison with SOTAs in PIAA

Following previous PIAA works [79, 80], We train a specific PIAA model for each of the 40 randomly selected annotators. The training set comprised 10 and 100 randomly chosen images per user, for the 10shot and 100shot tasks, respectively. Fifty additional images served as the test set. To mitigate the impact of random data selection, the train-test process is repeated 10 times across (randomly) different target users.

The average and standard deviation of PLCC/SRCC values on all test objects are reported in Table 4. Obviously, our AesMamba-P achieves the best performance, in both 10-shot and 100-shot task scenarios, according to the average PLCC and SRCC values. PARA* [79] uses one single attribute of users, i.e. artistic experience (*art.*), photographic experience (*pht.*), and personality trait (*psn.*). In contrast to PARA*, AesMamba-P gains an *absolute* improvement of 1.1-1.3 percents in PLCC, and of 1.8 percents in SRCC, for both 10shot and 100shot tasks. Such superiority demonstrates the significance of using users' multiple attributes for PIAA. Besides, AesMamba-P gains an *absolute* superiority of 0.3 and 0.6 percent improvement in PLCC and SRCC, respectively, over previous SOTA method, i.e. AFF-PIAA [92]. The possible reason of our performance fluctuation might be the few-shot learning settings. In other words, we consider users' multiple attributes, but there are only a few training samples. Fig. 1 shows the box-plot of SRCC values across the 10 times of random repeats. Inspiringly, AesMamba-P achieves significant superiority over existing PIAA methods, during most repetitions.

## 4.5 Ablation Study

In this section, we conduct a series of ablation studies, to analyze the major modules in AesMamba.

*4.5.1 Analysis of Multitask Balanced Adaptation (MBA).* In this part, we analyze the effectiveness of MBA, including the adapter for task adaptation, and the Bal-CE loss ($\mathcal{L}_{Bal-CE}$) for scale categorization (Section 3.4.2). To this end, we build several model variants of

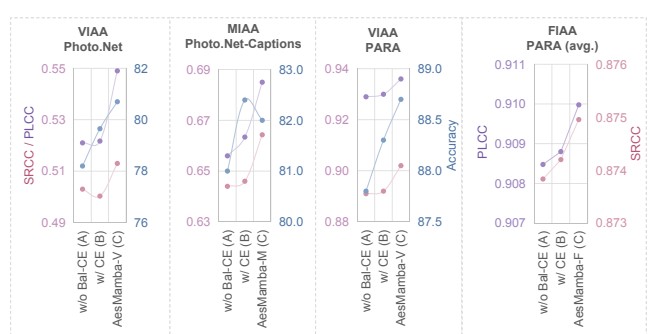

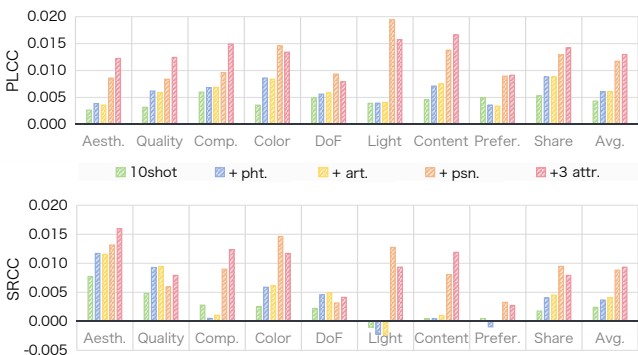

**Figure 6: Ablation study of _Multitask Balanced Adaptation_ (MBA) in diverse IAA tasks.**

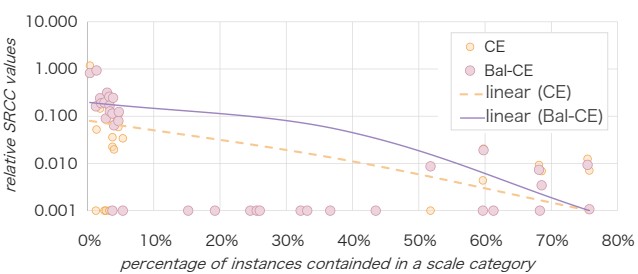

**Figure 7: The relative changes of SRCC (_logarithmic_ y-axis) vs. the percent of each scale category (x-axis), across all the attributes on PARA. "Linear" indicates the fitted trend line.**

**Figure 8: Impact of MCP for PIAA (10shot), on PARA. Each bar shows the change of PLCC/SRCC, compared to the PIAA performance of the initial GIAA model, AesMamba-V.**

AesMamba, and compare them in diverse IAA tasks, following the same experimental settings. The model variants include:

- Model-A: VMamba-T + task adaptation + $\mathcal{L}_{\text{MSE}}/\mathcal{L}_{\text{EMD}}$;
- Model-B: Model-A + categorization $\mathcal{L}_{\text{CE}}$;
- Model-C (full): Model-A + categorization $\mathcal{L}_{\text{Bal-CE}}$.

**Impact on diverse IAA tasks.** As shown in Fig. 6,Model-A, Model-B gains higher SRCC values across all the tasks, and boots the Accuracy and PLCC in most cases. This superiority indicates the potential of the auxiliary scale categorization task for performance boost. Besides, Model-C consistently achieves the best performance with significant superiority, across all the tasks. Such distinct superiority demonstrates the imperative role of balanced learning, for evaluating long-tail data, in the IAA task.

**Impact on each scale category in FIAA.** We further statistically evaluate the impact of the auxiliary categorization task on each scale category in the FIAA task. For each scale category of an attribute, we calculate the performance indices using the corresponding subset of instances. Fig. 7 visualizes the scatter plots of the relative change in SRCC values v.s. the percent of each category. Obviously, the aesthetic categorization task significantly boosts the performance for most few-shot categories, without significant decease for the other categories. Besides, the Bal-CE loss leads to better performance than the CE loss. Such observations demonstrate our motivation of using the auxiliary categorization strategy, for solving the challenge of long-tail data distribution in IAA.

### 4.5.2 Analysis of Subject Multi-character Prompting (SMP).
To verify the effectiveness of SMP, we build model variants of AesMamba-P by using none personal attribute (`10shot`), or one user's attribute (`+pht.`, `+art.`, or `+psn.`), or all three attributes (`+3 attr.`). We use the initial GIAA model, AesMamba-V (without fine-tuning), as the benchmark, and compare all the model variants to it. Fig. 8 shows the comparative results. In general, all the AesMamba-P model variants outperform AesMamba-V, demonstrating the significance of user-specific fine-tuning (`10shot`) and the use of personal attributes. In addition, among the three types of user information, personality traits (`+psn.`) play a pivotal role in PIAA performance improvement [79]. Finally, using all the three attributes (`+3 attr.`, i.e. AesMamba-P) achieves the best overall performance. Such observations demonstrate the effectiveness of our design of MCP, as well as the multimodal fusion module in integrating user information.

## 5 CONCLUSIONS

In this paper, we propose a universal IAA framework, AesMamba, that can be applied to diverse IAA tasks. The experimental results, across numerous datasets, demonstrate that AesMamba can precisely predict aesthetic and multi-attribute evaluations, based on visual or multimodal inputs. Besides, the proposed multitask balanced learning module, boosts the performance on tail instances; and the character prompting strategy significantly boost the PIAA performance. In the future, we will explore efficient and multimodal Mamba models, to further boost the IAA performance. Besides, it is meaningful to explore multimodal FIAA and efficient PIAA paradigms, for practical applications.

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
