# OpenReview forum: "AesMamba: Universal Image Aesthetic Assessment with State Space Models"
_acmmm.org/ACMMM/2024/Conference — MM2024 Oral_

### Official Review · Reviewer_prRf · 2024-05-23

**Rating:** 4
**Confidence:** 3

**Summary:**

In the paper, the authors propose a modular image aesthetic assessment (IAA) framework that can be applied to diverse IAA tasks through a flexible combination of modules in this framework. The framework can predict aesthetic and multi-attribute evaluations based on visual or multimodal inputs. In the personal image aesthetic assessment task, the authors introduce a character prompting strategy to boost the performance. In comparison with state-of-the-art methods, the proposed models achieve superior competitive performance on IAA tasks.

**Strengths:**

1. The idea of proposing a universal IAA framework with modular design for all IAA tasks is interesting.
2. The theoretical approach is relatively sufficient. The authors propose to use Visual Mamba for efficiently achieving both global and local effective reception fields.
3. Results are competitive. Comparison with other state-of-the-art methods shows that the proposed models obtain highly competitive performance.

**Limitations:**

1. The authors summarize five contributions in the paper. However, a universal IAA framework (the first contribution) includes Visual Mamba (the second contribution) a novel PIAA pipeline (the third contribution), and a Multitask Balanced Adaption (the fourth contribution). The summary of contributions is repetitive.
2. In the General IAA section of Related Works, the authors mention that previous multimodal general image aesthetic assessment (MIAA) requires a large amount of pre-trained data and huge computational resources. As far as I know, the proposed model in the paper also needs a corresponding dataset to pretrain or finetune. This is similar to previous works.
3. Some figures and descriptions are not enough detailed. In the (b) VSS Block of Figure 4, the author does not explain the calculation operation for the two branches. In the (c) SS2D of Figure 4, what does embedding in the S6 Block represent? How the four feature maps are merged to form a new image in the scan merge stage? The authors do not explain the meaning of ‘pht.’, ‘art.’ and ‘psn.’ in Figure 8.
4. The formula description is unclear. Equation 5 does not explain the meaning of z.
5. The design of subject multi-character prompts (SMP) relies on the specific dataset. If personal image aesthetic assessment datasets do not contain the corresponding attributes, this prompt template can not be used.
6. Experimental results presented in Section 4 are weak. A few methods of Visual GIAA are listed to compare with the proposed model on the PARA and Photo.Net database. At the same time, the performance of the proposed model is not outstanding and some more advanced methods such as AesCLIP have been released. Furthermore, the performance improvement of fine-grained image aesthetic assessment (FIAA) on the PARA dataset is not remarkable.
7. Results of the Ablation Study on diverse IAA tasks are weak. The AVA database is currently one of the largest and most widely used aesthetic databases. The ablation study of visual image aesthetic assessment (VIAA) and multimodal image aesthetic assessment (MIAA) should be operated on the AVA database instead of the Photo Net.
8.  The authors should provide source code to enhance the replicability of the proposed method.

**Suitability:**

3

---

### Official Review · Reviewer_w99M · 2024-05-24

**Rating:** 3
**Confidence:** 4

**Summary:**

The paper utilizes the Visual State Space Model (VMamba) instead of CNNs or ViTs for learning comprehensive aesthetic-related representations due to VMamba's efficient global and local receptive fields. A modal-adaptive module then integrates these representations based on input type. The Multitask Balanced Adaptation (MBA) module enhances task-specific features, focusing on tail instances. This work, AesMamba, is adaptable to various IAA tasks through its flexible approach.

**Strengths:**

1. This paper proposes a general framework to address multiple IAA tasks and introduces the use of Mamba in IAA for the first time.
2. Experimental results demonstrate that this method performs well.

**Limitations:**

1. the title "AesMamba" may be debatable since VManba is only used as the backbone.
2. It is recommended to investigate whether the proposed MBA and MAI modules can adapt to other backbones, such as ViT, by adding comparative experiments to illustrate the model's performance on different backbones.
3. The experimental details are unclear, particularly whether each task's model was trained independently or jointly during model training. If jointly, VMamba's weights might incorporate text information, leading to an unfair comparison.
4. The paper's writing requires significant improvement, with numerous grammatical errors and coherence issues. For instance, variants like AesMamba-F are mentioned only in the experimental section without clear reference in the methodology.

**Suitability:**

3

---

### Official Review · Reviewer_b8E4 · 2024-05-24

**Rating:** 6
**Confidence:** 3

**Summary:**

This paper showcase an universal IAA framewrk namely AesMamba and it is applicable for multimple IAA settings by combining proposed modules. They also provide a novel PIAA pipeline named subject multi-character prompts to encode subject characters into text embedding for multimodal learning, which is novel. They further take imbalance problem into consideration and learn task-adaptive representation with emphasis on tail class. The entire pipeline is novel and this work covers most important research problems in IAA area.

**Strengths:**

1. Technical novelty is good with novel task setting and achieving SOTA performance with proposed framework.
2. Experiment design is good and well-covered, including most IAA research tasks and the experimental results are so far the most completely organized, which is a solid contribution to the research field.
3. They recognize the key problem in IAA research, i.e. the natural imbalanced data. They design three model variants to see its influence.
4. Choose to solve imbalance problem in IAA research shows that the authors have good taste. When facing with continuous space imbalance problem, discretization is a good way out by jointly considering features of imbalanced regression.
5. Taking user information as text prompt is a novel way of thinking and a new task setting.

**Limitations:**

Writing quality should be improved a bit. e.g. in section 3.6, "In some benchmark dataset...", which should be more precise by given the name directly.

In addition, some short forms are not quite clear. e.g. the MCP, which may refer to multi-character prompts (MCP), while the actual short form in this paper is "subject multi-character prompts (SMP)". I am not sure if they refer to same thing, but please do double check your manuscript carefully.

Last but not least, it would be great if you can add limitation discussion which may bring new ideas to this research area.

**Suitability:**

3

---

### Official Review · Reviewer_o16M · 2024-06-04

**Rating:** 4
**Confidence:** 2

**Summary:**

The paper presents a novel modular framework called AesMamba for Image Aesthetic Assessment (IAA). AesMamba leverages Visual State Space Models (VMamba) for learning comprehensive visual representations and introduces several innovative components such as a modal-adaptive module, Multitask Balanced Adaptation (MBA) module, and a prompting strategy for personalized IAA. The proposed framework demonstrates superior or competitive performance across a range of IAA tasks and datasets.

**Strengths:**

The use of State Space Models (SSMs) in the IAA domain is novel and promising, offering efficient learning of both local and global visual representations.
The introduction of the MBA module to address task-level and instance-level imbalances is a significant contribution.

**Limitations:**

Some sections of the paper, particularly those describing the technical implementation of the VMamba and MBA modules, are dense and could benefit from clearer explanations and more visual aids.
The paper predominantly focuses on PLCC and SRCC metrics. Including a broader range of evaluation metrics could provide a more comprehensive assessment of model performance.

**Suitability:**

3

---

### Meta-Review · Area_Chair_hzHR · 2024-06-30

**Recommendation:** Accept (Oral)
**Confidence:** 4

**Metareview:**

This paper proposes a modular IAA framework for universal IAA.
The validity of the proposed work is well-demonstrated by the experiments.
On the other hand, this paper would benefit from revising the tachnical details part.

All the reviewers are positive about this paper. I would also vote for the acceptance of this paper.